# Beneficial Effects of the Very-Low-Calorie Ketogenic Diet on the Symptoms of Male Accessory Gland Inflammation

**DOI:** 10.3390/nu14051081

**Published:** 2022-03-04

**Authors:** Rosita A. Condorelli, Antonio Aversa, Livia Basile, Rossella Cannarella, Laura M. Mongioì, Laura Cimino, Sarah Perelli, Massimiliano Caprio, Sebastiano Cimino, Aldo E. Calogero, Sandro La Vignera

**Affiliations:** 1Department of Clinical and Experimental Medicine, University of Catania, 95123 Catania, Italy; rosita.condorelli@unict.it (R.A.C.); rossella.cannarella@phd.unict.it (R.C.); lauramongioi@hotmail.it (L.M.M.); lauracimino@hotmail.it (L.C.); sarah.perelli@libero.it (S.P.); sandrolavignera@unict.it (S.L.V.); 2Department of Experimental and Clinical Medicine, University Magna Graecia of Catanzaro, 88100 Catanzaro, Italy; aversa@unicz.it; 3Department of Chemical Sciences, University of Catania, 95123 Catania, Italy; livia.basile@unict.it; 4Laboratory of Cardiovascular Endocrinology, IRCCS San Raffaele Pisana, 00163 Rome, Italy; massimiliano.caprio@uniroma5.it; 5Department of Human Sciences and Promotion of the Quality of Life, San Raffaele Roma Open University, 00166 Rome, Italy; 6Section of Urology, Department of Surgery, University of Catania, 95123 Catania, Italy; ciminonello@hotmail.com

**Keywords:** VLCKD, very-low-calorie ketogenic diet, MAGI, male accessory gland infection, Mediterranean diet, urinary symptoms, quality of life, sexual dysfunction, ejaculatory pain

## Abstract

**Introduction.** Obesity exposes individuals to the risk of chronic inflammation of the prostate gland. **Aim and design of the study.** A longitudinal clinical study was conducted on selected overweight/obese patients with male accessory gland inflammation (MAGI) to evaluate the effects of body weight loss on their urogenital symptoms. **Materials and methods.** One hundred patients were selected and assigned to two groups undergoing two different nutritional programs. The first group (*n* = 50) started a Mediterranean diet (MedDiet) and the second (*n* = 50) a very-low-calorie ketogenic diet (VLCKD). Before and after three months on the diet, each patient was evaluated for body weight, waist circumference, and MAGI symptoms. The MAGI was assessed using the Structured Interview about MAGI (SI-MAGI), a questionnaire previously designed to assess the symptoms of MAGI. The questionnaire explores four domains, including urinary symptoms, ejaculatory pain or discomfort, sexual dysfunction, and impaired quality of life. Finally, in the two groups, the frequency of an α-blocker used to treat urinary tract symptoms was also evaluated. **Results.** Patients on MedDiet experienced significant amelioration in urinary symptoms and quality of life. Patients under VLCKD reported not only significant improvement of the same parameters, but also in ejaculatory pain/discomfort and sexual dysfunction. Finally, the percentage of patients on VLCKD taking the α-blocker decreased significantly. Moreover, patients under VLCKD showed a greater loss of body weight than those following the MedDiet. **Discussion.** The results of this study support the effectiveness of VLCKD in improving the symptoms of patients with MAGI. This improvement involved all of the domains of the SI-MAGI questionnaire and became manifest in a relatively short time. We suggest that a ketogenic nutritional approach can be used in overweight/obese patients with MAGI.

## 1. Introduction

Male accessory gland infections/inflammations (MAGI) are a frequent reason for clinical consultation in reproductive and sexual medicine. Their frequency is very heterogeneous due to the different applications of the diagnostic criteria, but it is estimated to range from 2 to 18% [1,2]. Initially, MAGI were observed among patients with male infertility; subsequently, they have been described with increased frequency in patients with sexual dysfunction [1].

The factors recognized as contributing to MAGI development include sexual promiscuity, irritable bowel, urethral catheterization, and morphostructural alterations of the prostate gland [2]. Metabolic factors, particularly obesity, expose individuals to the risk of prostate inflammation. This is supported by both clinical and experimental evidence [3]. Clinically, MAGI can be classified into uncomplicated forms when they only involve the prostate (prostatitis), and complicated forms when the inflammatory process also involves the seminal vesicles (prostate vesiculitis) and the epididymis (prostate vesicle epididymitis). From a microbiological point of view, MAGI are classifiable as microbial and inflammatory forms [1].

The choice of MAGI treatment is made according to the symptoms or needs of the patient. MAGI may require treatment of signs (alterations in sperm parameters) and be deserving of pharmacological treatment with a sequential strategy (antibiotic therapy, followed by anti-inflammatory or corticosteroid therapy, followed by antioxidant therapy), or treatment for urinary symptoms, ejaculatory pain and/or discomfort, and/or sexual dysfunction. In the latter cases, α-blockers and type V phosphodiesterase inhibitors are generally used [1,4]. However, the symptoms of patients with MAGI are not easy to assess because they can be mild in intensity and due to the frequent chronic course of MAGI. For this reason, we developed a questionnaire that explores four functional domains and, using a few targeted questions, can bring out the main symptoms by differentiating between urinary symptoms, pelvic pain, and sexual disorders/discomfort [5].

Weight reduction is associated with symptom improvement in men suffering from prostatitis [6], but there is little evidence on the most appropriate nutritional choices for symptom treatment in these patients [7]. The very-low-calorie ketogenic diet (VLCKD) mimics fasting by markedly reducing carbohydrate intake (<30 g/day), and concomitantly increasing the intake of fat and protein [8]. This induces the production of ketone bodies, such as D-3-β-hydroxybutyrate, acetoacetate, and acetone. These are anorexigenic molecules produced in the mitochondria of hepatocytes, which reduce cerebral neuropeptide Y, ghrelin, and maintain the cholecystokinin response to a meal. These molecular mechanisms lead to a reduction in both perceived hunger and food intake, which explains the efficacy and tolerability of VLCKD [9]. Ketone bodies appear to have a beneficial effect on different tissues, and the Italian Society of Endocrinology recommends the use of VLCKD for several obesity-related complications, such as type II diabetes mellitus, hypertension, or dyslipidemia [10]. It is currently unknown whether ketone bodies have a positive effect on prostate tissue, although recent evidence suggests that low-carbohydrate diets could be used to reduce the progression of prostate cancer [11]. So far, no evidence is available on the effects of VLCKD on prostatitis.

Therefore, we aimed to assess the impact of the VLCKD or the Mediterranean diet (MedDiet) on the symptom profile (voiding disorders, and ejaculatory and sexual disorders) in selected overweight or mildly obese patients with MAGI.

## 2. Patients and Methods

### 2.1. Patient Selection

One hundred patients with inflammatory bilateral prostate vesicular epididymitis (PVE) associated with overweight or mild obesity were enrolled (Table 1). The patients were divided in two groups: 50 patients underwent a MedDiet, and the remaining 50 patients undertook the VLCKD protocol. The scores of a specific questionnaire previously devised for patients with MAGI [5], administered before and after three months of the diet, were collected. The nutritional protocol was chosen by patients according to their nutritional and organizational preferences during their initial clinical consultations.

### 2.2. Exclusion Criteria

Patients with the following diseases were not enrolled in this study: microbial MAGI, papillomavirus infection, moderate or severe obesity, hypogonadism, previous history of erectile dysfunction (ED), life-long premature ejaculation, varicocele, spinal pathology, peripheral neuropathy, bladder disease, kidney stones, and mood alterations. Patients who had undergone drug treatment for MAGI in the previous six months were excluded from the study, except patients taking alfuzosin therapy (the only drug of this class not associated with retrograde ejaculation) for the treatment of bladder voiding disorders. Criteria already adopted in the literature were used for the diagnosis of MAGI [1].

### 2.3. Nutritional Programs

#### 2.3.1. Very-Low-Calorie Ketogenic Diet

The VLCKD protocol was based on a sharp reduction in the intake of carbohydrate (<30 g/day), and an assumption of ~44% of fat and of ~43% of protein. In this study, we gave patients 1.4–1.5 g of protein per kg of the ideal body weight daily. The dietary protocol included the following phases [12]:Phase I involved the replacement of natural proteins with five substitutive meals. During lunch and dinner, the patients were allowed to eat low glycemic index vegetables. The total amount of calories was 600–800 kcal/day (VLCKD).During phase II, two different programs could be chosen. The first allowed natural protein food (meat, eggs, or fish) for lunch or dinner. The second option consisted of taking protein preparations during breakfast and snacks and replacing both main meals with natural proteins. The total amount of calories was 800–1000 kcal/day (low-calorie ketogenic diet). In both programs, only low glycemic index vegetables were allowed. The first and second phases lasted for 12 weeks, during which ketosis was maintained. Micronutrients, consisting of vitamins, minerals, and omega-3 fatty acids, were recommended.Phase III consisted of the gradual reintroduction of carbohydrates. Foods and vegetables with a higher glycemic index were also used in this phase to replace the protein preparations. The latter were used only for breakfast and a snack, while the other snack consisted of fruit. In this phase, the patients were advised to undertake a physical activity program of at least 10 min daily. The total amount of calories was 1200–1500 kcal/day (low-calorie diet).Phases IV and V were characterized by the reintroduction of pasta or bread for lunch, cereals for breakfast or dinner, and legumes for lunch or dinner. The total amount of calories was 1500–2000 kcal/day.

In this study, the diet protocol was followed for at least 12 weeks, including phases I and II.

#### 2.3.2. Mediterranean Diet

The control group followed the Mediterranean diet, according to the following protocol:45–60% of carbohydrates, mainly complex (such as cereal starches);10–12% of proteins, corresponding to 0.9 g per kg of body weight;20–35% fat with less than 10% of saturated fats (mainly represented by animal products, with the exception of fish).

The number of calories introduced was established based on the calculation of the daily caloric requirement with a consequent reduction of 20% of the estimated intake. The principles of the Mediterranean diet used in this study were:(1)A high ratio of monounsaturated to saturated dietary lipids (mainly olive oil);(2)Moderate ethanol intake;(3)High consumption of legumes;(4)High consumption of unrefined grains, including bread;(5)High fruit consumption;(6)High consumption of vegetables;(7)Low consumption of meat and products derived from meat;(8)Moderate consumption of milk and dairy products.

### 2.4. Side Effects

During the three months of the nutritional protocol, the following side effects were observed through telematics contact with the patients for the evaluation of any study drop-out criteria: headache, dry mouth, dizziness, hypotension, visual abnormalities, hypoglycemia, halitosis, lethargy, constipation, diarrhea, hyperuricemia, vomiting/nausea, urolithiasis, hair loss, and gallbladder disease.

### 2.5. Physical Activity

The physical activity program proposed to the patients enrolled in this study was identical for the two groups. It was derived from the application of Karvonen’s formula: frequency of cardiac reserve (FCR) = maximum heart rate—basal heart frequency. In particular: THR = (maximum heart rate* − basal heart rate) × %intensity) + basal heart frequency, where the intensity was set at 70–80% for aerobic work. The dose of physical activity was 150 min of moderate-intensity aerobic activity weekly. The duration of the work session was 30 min. As practical advice, it was suggested that all patients practice physical activity while keeping their heart rate between 40 and 60% of their maximum heart rate. *Maximum heart rate: 220—age of the patient [13].

### 2.6. Questionnaire on Male Accessory Gland Inflammation

The questionnaire administered before and after the three months of the nutritional intervention was the Structured Interview on MAGI (SI-MAGI) previously developed by our research group for the anamnestic evaluation of patients with MAGI. It explores the following four domains: urinary symptoms, ejaculatory pain or discomfort, sexual dysfunction, and impairment of the quality of life (QoL) [5].

The protocol of the study is shown in Figure 1.

### 2.7. Ethical Approval

The protocol was supervised and approved by the internal Institutional Review Board of the Division of Endocrinology, Metabolic Diseases, and Nutrition, University-Teaching Hospital Policlinico “G. Rodolico—San Marco”, University of Catania (Catania, Italy), where the study was carried out. The principles enunciated by the Declaration of Helsinki were applied and strictly followed. We offered a full explanation of the study purpose to each participant, and written informed consent was signed by each of them.

### 2.8. Statistical Analysis

The results of the study are shown as the mean ± SD for normally distributed variables. The Shapiro–Wilk test was used to evaluate the distribution of each analyzed variable. The one-way analysis of variance (ANOVA), followed by the Tukey–Kramer post hoc test, used to evaluate the within-group differences. The Chi-squared test was used to analyze the difference in the percentage of men using α-lithics in the MedDiet and the VLCKD groups. The statistical analysis was performed using MedCalc Software Ltd. (Version 19.6—64 bit). The results were considered statistically significant for a *p*-value lower than 0.05.

## 3. Results

Patients in the MedDiet group were similar in age (26.4 ± 4.8 years) and height (170.5 ± 3.2 cm) to those of the group of MAGI patients undergoing the VLCKD (25.6 ± 4.6 years and 171.2 ± 3.3 cm, respectively). After the VLCKD, patients showed significant decreases in body weight, body mass index (BMI), and waist circumference compared to baseline and patients on the MedDiet (Table 1). The latter showed no significant differences in terms of body weight, BMI, and waist circumference after 3 months on the MedDiet. None of the patients discontinued the study due to the onset of side effects.

The evaluation of MAGI symptoms using the SI-MAGI questionnaire revealed that patients on the MedDiet had a significant reduction in the severity of urinary symptoms and amelioration of the QoL compared to the scores obtained from these patients at enrolment in the study (Figure 2, panels A,D).

Patients who practiced the VLCKD showed a significant reduction in the severity of the urinary symptoms, ejaculatory pain or discomfort, sexual dysfunction, and amelioration of the QoL, compared to the score obtained at enrolment by these patients. Furthermore, all of these scores improved significantly in this group of patients compared to those on the MedDiet (Figure 2A–D). Finally, the percentage of α-blocker users in the group of patients on the VLCKD decreased significantly compared to baseline and compared to patients on the MedDiet (Figure 3).

## 4. Discussion

This study demonstrated that a ketogenic nutritional approach is more effective on both body weight loss and MAGI symptoms than the Mediterranean diet. The greater efficacy of the VLCKD on body weight loss can be attributed to the greater practicality of the protocol, especially in young and daily active patients. The more recent scientific literature has highlighted the benefits of the ketogenic diet that can be now considered a well-established nutritional option in clinical practice for its efficacy and safety. A very recent study by Barrea and collaborators showed that the interruption of the dietary protocol is due, in the vast majority of cases, to its palatability or excessive costs. Side effects are very mild and easily managed by specialists with experience in this field [14].

This study was undertaken to assess the urogenital disorders in patients with MAGI on a ketogenic diet. Despite the relevant link between body weight and prostate enlargement and consequent symptoms, this is not a conventional aspect. In addition, in our clinical experience [12] on VLCKD nutritional protocol, we have given greater importance to different aspects including the recovery of hypothalamic–pituitary–testicular axis function in patients with metabolic hypogonadism and its protective effects on pancreatic beta cells in obese men.

The efficacy of both diet regimens, particularly the VLCKD, on the main symptoms of MAGI is certainly attributable to the benefit associated with weight loss on prostate inflammation. Indeed, obesity negatively affects prostate function through a series of well-known mechanisms, the main of which are hormonal imbalance (relative hyperestrogenism), increased lymphocyte infiltration of the prostate tissue, and increased adrenergic tone [15].

Dietary interventions for chronic diseases are useful therapeutic options for their health benefits, lack of unwanted side effects, and low cost compared with pharmaceuticals. However, dietary recommendations are still not used for the treatment of andrological disorders. To our knowledge, this is the first study that evaluated the benefits of diet on MAGI symptoms in selected patients with PVE. The results showed the efficacy of a weight-loss strategy, but the efficacy of the VLCKD was superior to that of the Mediterranean diet in the short observation period (three months). Unfortunately, very few studies have explored this aspect.

In 2013, Herati and colleagues administered a questionnaire to 286 patients with chronic prostatitis/chronic pelvic pain syndrome (CP/CPPS). The questionnaire listed several questions about the effects of food and drink on symptoms caused by CP/CPPS and questions about the frequency of consumption of food knew to flare up their symptoms. The results indicated 176 foods classified in the following categories: fruit, vegetables, meat, fish, and poultry; bread and cereals; sweets and snacks; drinks; ethnic and various food. A total of 43.5% of patients affirmed that specific foods and drinks could worsen the pain and the frequency and urgency of urination after their consumption, while 22.6% reported that some foods were able to relieve their symptoms. Spicy food was listed as the most irritating, followed by coffee, hot peppers, alcoholic beverages, tea, and chili. Interestingly, decaffeinated coffee was significantly less of a nuisance than caffeinated coffee. This suggests that caffeine may play a role in the exacerbation of symptoms. Indeed, other caffeinated foods have been mentioned as causing pain (e.g., tea, chocolate, soda). Another interesting finding was that patients often reported that foods containing hot pepper as an ingredient were capable of inducing pain. Hot peppers derive their spiciness from capsaicin, which increases rectal sensitivity in patients with irritable bowel syndrome [7].

A case-control Chinese study showed that among dietary habits, alcohol consumption and spicy food have a detrimental impact on CP, while the increase in the intake of water was a protective factor. The same study showed that both vegetables and meat-based diets are associated with a higher risk of CP [16].

It has long been recognized that diet can influence prostate health and enhances the benefits of traditional clinical practice. Diet is considered part of the treatment of various prostate diseases from benign prostate hyperplasia (BPH) to prostate cancer [17,18,19]. Kristal and colleagues found a correlation between a reduced risk of developing BPH and taking nutraceutical compounds, such as vitamin D, zinc, or lycopene [20]. The authors speculated a role for vitamin D in the RhoA/ROCK pathway inhibition, in the expression of cyclooxygenase-2 expression, and in the production of PGE2 in BPH stromal cells. The increased consumption of vitamin D from diet or supplementation resulted in a reduction in the risk of BPH. A double-blind randomized controlled trial (RCT) in patients with BPH and a prostate volume higher than 40 mL showed a significant decrease (−2.90%) in prostate volume after taking the vitamin D analog BXL628 (150 μg/day = 6000 IU) for 10 months [19]. In line with the available evidence, the prevalence and the severity of prostate diseases is lower in areas where a plant-based diet is prevalent. In contrast, the risk for BPH is higher in case of fat- and red meat-based diets or dietary regimens with low intake of proteins and vegetables [20]. Lagiou and colleagues found that consumption of fruits with high levels of β-carotene, lutein, or vitamin C had positive effects on the outcomes of BPH in a cohort of 420 patients [21]. Furthermore, the pro-inflammatory activity of this type of high-fat diet can explain its association with the increased risk for BPH development, since this type of food can induce prostate tissue inflammation, hypoxia, and remodeling. It seems evident that different types and compositions of fat can influence the development of BPH. De Ribeiro and colleagues suggested that flaxseed reduced the proliferation of prostate epithelial cells of rats with BPH [22]. According to the evidence from an RCT carried out in 87 patients with BPH, the flaxseed extract secoisolariciresinol diglucoside (SDG) at the dose of 300 or 600 mg/day reduced the international prostate symptom score (IPSS), increased the QoL score, and improved the grade of lower urinary tract symptoms (LUTS) (from moderate/severe to mild). The efficacy of flaxseeds was also compared to that of α1A- blockers and 5α-reductase inhibitors [23].

In addition, a prospective study showed that patients with high BMI and total and abdominal fat had a higher chance of developing LUTS [24]. On the other hand, modifications of the lifestyle (such as diet and exercise) can positively affect LUTS. Adedeji and colleagues also described the presence of a positive correlation between diet and metabolic syndrome, resulting in an increased risk for LUTS [25].

ED can also be impacted by diet. Recently, Lu and colleagues described the existence of a relationship between ED and a plant-based diet in a Chinese study of 184 patients classified into two groups: patients with or without ED (*n* = 92 each). At the time of enrolment, patients in both groups had similar characteristics (age, age of partners, length of relationship, monthly income, education level, frequency of sexual intercourse, occupation, and place of residence). However, they differed in terms of lifestyle (such as exercise, smoking, and alcohol intake). The plant diet index (PDI) and the healthy plant diet index (hPDI) of patients without ED were significantly higher than in those with ED. Multivariate analysis showed that the presence of ED reduces the bioavailability of nitric oxide, as well as the PDI and hPDI levels, and improves BMI, metabolic syndrome, and E-selectin levels. Furthermore, both the PDI and hPDI values and the international erectile function index (IIEF-5) scores increased in the ED group [26]. Previous studies have focused on the association between other types of diet and erectile function. For example, Ramìrez and colleagues indicated that a low prevalence of ED can be found in patients adhering to the Mediterranean diet, and consuming fish, vegetables, fruit, whole grains, and nuts [27,28]. This diet has anti-inflammatory properties, since it is rich in flavonoids [29,30]. Accordingly, a decreased prevalence of ED was reported when nuts were consumed more than twice a week and vegetables more than once a day. Similarly, according to the data of a prospective study performed on 555 type 2 diabetic patients, randomly assigned to the Mediterranean or high-fat diet, the Mediterranean diet seemed to be associated with a reduction in ED risk [27,31,32]. In addition, adherence to low-fat diets and consequent loss of weight in obese patients with ED can lead to an improvement of the IIEF-5 score [33,34].

Recently, some evidence indicated that both no-carbohydrate ketogenic (NCKD) and low-carbohydrate diets could slow the progression of prostate cancer. Accordingly, in mouse models with prostate cancer, the administration of these diets caused a slower growth of the tumor compared to those on a high fat diet [11]. Additionally, the use of a ketogenic diet improved prostate hyperplasia in rats, and could represent a cheaper and alterative therapy for BPH [35]. Although the evidence assessing the impact of ketogenic diet on prostate cancer in humans is still limited, recently, a role of ketones for their anti-cachectic properties has been suggested [36]. Currently, two different trials are evaluating the effects of an NCDK diet in prostate cancer patients under active surveillance (NCT02194516, NCT03679260). Similarly, Tulipan and colleagues studied the effect of the NCKD diet on the QoL of several cancer patients via an online questionnaire. In this study, breast cancer, with a prevalence of 44% among the participants, was the most frequently identified type of tumor, followed by cervical, prostate, colorectal cancers, and melanoma. Half of the participants (51%) followed a ketogenic diet during cancer therapy. In detail, a low carbohydrate/high-fat diet was followed by 21% of the participants, and a low glycemic index diet by 12%. Interestingly, the results suggested improved QoL in more than two-thirds of the study participants. Furthermore, social support, affinity with cooking, propensity to seek out new recipes, and a general preference for fatty foods have been found to influence the implementation of ketogenic diets in cancer patients. While the primary source of calories is corn oil in high-fat diet models, lard and milk fat are used in the NCKD and low-carbohydrate models [37]. Corn oil is mainly composed of omega-6 fatty acids and has an omega-3 to omega-6 ratio of 1:50 [38]. Lard has an omega-3 to omega-6 ratio of 1:10. Therefore, the NCKD provides an intake of omega-3 five times higher than a high-fat diet. Considering the lower omega-3 intake, a high-fat diet could induce inflammation and be more oncogenic [39].

To our knowledge, this is the first clinical study demonstrating the protective effects of weight loss on the symptom complex of patients with MAGI. This improvement is associated with a reduction in the need for α-blocker therapy for the management of urinary symptoms. In particular, the ketogenic diet appears to be a more effective option than the Mediterranean diet. Further studies deserve to be conducted on a larger number of patients to evaluate the possible synergy with the drugs commonly used for their treatment, and to explore the potential impact on the sperm parameters of these patients.

## Figures and Tables

**Figure 1 nutrients-14-01081-f001:**
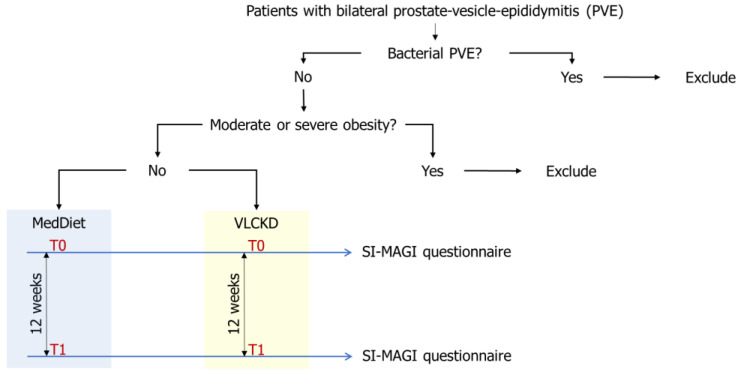
Protocol of the study. Patients with inflammatory prostate vesicle epididymitis who were overweight or mildly obese underwent a Mediterranean diet (MedDiet) or a very-low-calorie ketogenic diet (VLCKD) for at least 12 weeks. The Structured Interview on Male Accessory Gland Inflammation (SI-MAGI) questionnaire was administered to both groups before the start (T0) and after 12 weeks (T1) of the diet. Patients with bacterial PVE or moderate to severe obesity were excluded.

**Figure 2 nutrients-14-01081-f002:**
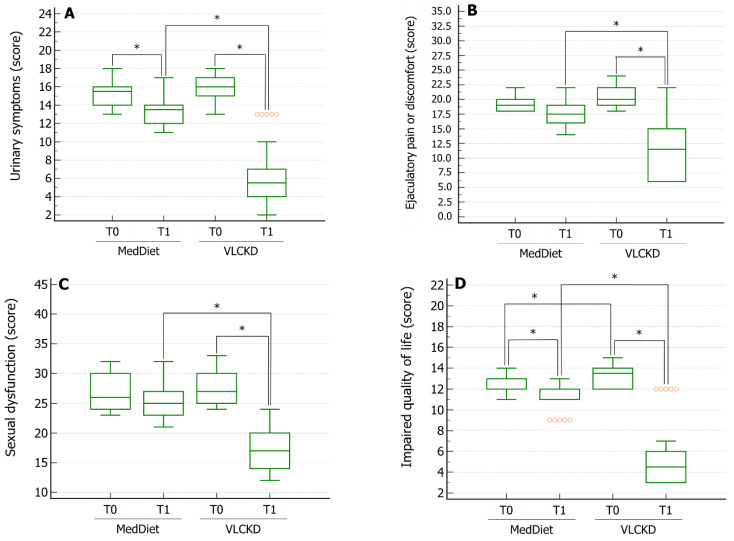
Scores were obtained by administering the Structured Interview on Male Accessory Gland Inflammation (MAGI) on urinary symptoms (Panel **A**), ejaculatory pain or discomfort (Panel **B**), sexual dysfunction (Panel **C**), and impaired quality of life (Panel **D**) in patients with MAGI before (T0) and after three months (T1) on a Mediterranean diet (MedDiet) or very-low-calorie ketogenic diet (VLCKD). * *p* < 0.05 by one-way analysis of variance followed by the Tukey–Kramer post hoc test.

**Figure 3 nutrients-14-01081-f003:**
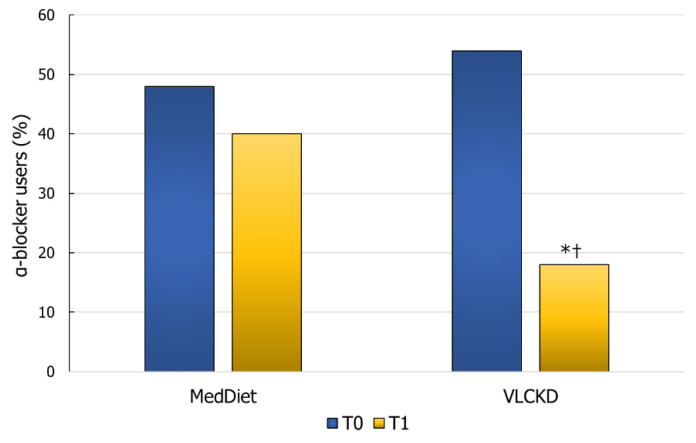
Percentage of α-blocker users for the treatment of urinary symptoms in patients with male accessory gland inflammation before (T0) and after three months (T1) on a Mediterranean diet (MedDiet) or very-low-calorie ketogenic diet (VLCKD). * *p* < 0.05 vs. VLCKD T0; ^†^
*p* < 0.05 vs. Med-Diet T1 (Chi-squared test).

**Table 1 nutrients-14-01081-t001:** Bodyweight, body mass index (BMI), and waist circumference of patients with male accessory gland inflammation before (T0) and after three months (T1) on a Mediterranean diet (MedDiet) or very-low-calorie ketogenic diet (VLCKD).

	MedDiet		VLCKD	
	T0	T1	T0	T1
Body weight (kg)	89.3 ± 5.7	87.0 ± 6.4	90.2 ± 5.6	79.2 ± 5.1 *^,†^
Body mass index (kg/m^2^)	30.8 ± 2.3	30.0 ± 2.6	30.8 ± 2.0	27.1 ± 1.8 *^,†^
Waist circumference (cm)	104.1 ± 9.2	102.5 ± 6.3	105.2 ± 5.2	96.3 ± 10.5 *^,†^

* *p* < 0.05 vs. VLCKD T0; ^†^
*p* < 0.05 vs. MedDiet T1 (one-way analysis of variance followed by the Tukey–Kramer post hoc test).

## Data Availability

The raw data are available upon request to the corresponding author.

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
