# Peer review of "Beneficial Effects of the Very-Low-Calorie Ketogenic Diet on the Symptoms of Male Accessory Gland Inflammation"

_nutrients, 2022, doi:10.3390/nu14051081_

Round 1

Reviewer 1 Report

Dear Authors,

thank you for an interesting article about ketogenic diet. It shows how nutrition is important for a healthy life. I think there are small ways to improve the manuscript. 

  1. More information about ketogenic diet and how it works at the molecular level. I think it would help readers to understand that beneficial effect of the diet used in the study.
  2. According to me, the schematic picture of the study protocol would present nicely what was done during the study.

Author Response

Answers to Reviewer #1 comments

Manuscript ID nutrients-1601223 Revised

Comment 1. Thank you for an interesting article about ketogenic diet. It shows how nutrition is important for a healthy life.

Answer to Comment 1. We appreciated the time that you spent in reviewing our manuscript as well as your constructive criticisms.

Comment 2. I think there are small ways to improve the manuscript. More information about ketogenic diet and how it works at the molecular level. I think it would help readers to understand that beneficial effect of the diet used in the study.

Answer to Comment 2. Thank you for this comment. More information was added, please see lines 69-82, which reads as follows:

The very-low-calorie ketogenic diet (VLCKD) is a nutritional program that mimics fasting by markedly reducing carbohydrate intake (<30 g/day), with a relative increase in fat and protein intake [8]. This induces the production of ketone bodies, such as D-3-β-hydroxybutyrate, acetoacetate, and acetone. These are anorexigenic molecules produced in the mitochondria of hepatocytes, which reduce cerebral neuropeptide Y, ghrelin, and maintain the cholecystokinin response to a meal. These molecular mechanisms lead to a reduction in perceived hunger and food intake, which explains the efficacy and tolerability of VLCKD [9]. Ketone bodies appear to have a beneficial effect on different tissues and the Italian Society of Endocrinology recommends the use of VLCKD for the treatment of various complications related to obesity, such as type II diabetes mellitus, hypertension, or dyslipidemia [10]. It is currently unknown whether ketone bodies have a positive effect on prostate tissue, although recent evidence suggests that low-carbohydrate diets could be used to slow the progression of prostate cancer [11]. So far, no evidence is available on the effects of VLCKD on prostatitis.

Comment 3. According to me, the schematic picture of the study protocol would present nicely what was done during the study.

Answer to Comment 3. We agree with this comment and added a figure explaining the study protocol (please see the new Figure 1).

Reviewer 2 Report

1.- In “line 115” you comment that in this phase (third phase), at least 10 min of physical activity  per day were suggested.

However, in “line 145”  you refer to a planned physical activity, which does not coincide with what you suggest in line 115.

Please clarify this point.

2.- (line 216) The more recent scientific literature has highlighted the benefits of the ketogenic diet that cab be now considered

( Correct English Grammar: (can)

Author Response

Answers to Reviewer #2 comments

Manuscript ID nutrients-1601223 Revised

Comment 1. In “line 115” you comment that in this phase (third phase), at least 10 min of physical activity per day were suggested. However, in “line 145” you refer to a planned physical activity, which does not coincide with what you suggest in line 115. Please clarify this point.

Answer to comment 1. During phase 3, patients were advised an at least 10-minute-long physical activity program, daily. We clarified this in lines 130-132, which reads as follows: “In this phase, patients were advised to undertake a physical activity program of at least 10 min daily”. We deleted the word “planned” in line 145. Physical activity was suggested to both MedDiet and VLCKD groups (please see paragraph “Physical activity”.

Comment 2. .- (line 216) The more recent scientific literature has highlighted the benefits of the ketogenic diet that cab be now considered. Correct English Grammar: (can)

Answer to comment 2. Corrected, thank you.